# Plasma and Colostrum Selenium Statuses of Properly Supplemented Belgian Blue Cows on Commercial Farms and Their Relationship with Sources of Dietary Selenium and Blood Biomarkers

**Pauline Delhez** , **Émilie Knapp, Barbara Pirard, Marceau Gauthier, Anne-Sophie Rao, Christian Hanzen** and **Léonard Theron** *

RumeXperts, 4317 Faimes, Belgium
* Correspondence: leonardtheron@rumexpert.vet

**Abstract:** Selenium (Se) is an essential trace element for the health and immunity of cattle. Double-muscled Belgian Blue cows are well known to be prone to nutritional deficiencies. Colostrum Se level is also a key factor to promote immunoglobulin intake in young calves. The main objectives of this study were to assess (1) the plasma and colostrum Se statuses of properly supplemented Belgian Blue cows on commercial farms and (2) the relationship between Se concentrations in plasma and colostrum. The secondary objectives were to assess relationships between plasma or colostrum Se concentrations and dietary Se supplementation as well as blood biomarkers. Blood and colostrum samples were collected from 49 Belgian Blue cows on five commercial farms in Belgium. They received five different rations with Se supplementation ranging from 0.5 to 2 ppm, including 20% to 83% in organic form. Results showed that the average Se concentration was $90 \pm 15$ µg/L in plasma and $79 \pm 26$ µg/L in colostrum, consistent with previous studies on well-supplemented cows. No relationship was observed between Se concentrations in plasma and colostrum, suggesting that colostrum Se testing would be a complementary indicator for improving calf Se supplementation. Relationships between plasma or colostrum Se concentrations and dietary Se or blood biomarkers emphasized the complexity of Se metabolism in observational studies under field conditions.

**Keywords:** cattle; Belgian Blue; beef; colostrum; selenium

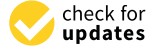



## 1. Introduction

Selenium (Se) is an essential trace element for ruminants. It plays multiple roles in the health and performance of cattle, for instance, in the improvement of the antioxidant defense, immune system, fertility, milk quality, growth, etc. [1,2]. In the newborn calf, Se is also important for immunoglobulin absorption from colostrum [3,4]. Thus, Se deficiency has economically significant impacts on farms, underlying the importance of proper supplementation and monitoring of the herd status [5,6]. The most common form of supplementation is through the addition of Se in the diet of the pregnant cow. The last months of gestation and the postpartum period are the critical periods in terms of Se availability for the upcoming calf through placental, colostrum and milk transfer [7]. However, the relationship between Se supplementation and Se concentration in the blood, or in colostrum, is not always straightforward. Indeed, it depends on complex interactions in the Se absorption, stockage capacity and also the physiological state of the animal (age, production, health status, protein status, etc. [8,9]). The form of the dietary Se has also an impact on Se concentration in the blood and in colostrum. For example, several studies have shown that the addition of Se in organic form in the feed leads to a more pronounced increase in the Se content in blood and colostrum compared to supplementation with Se in an inorganic form, although these effects are complex [10,11].

Se in beef cattle has been studied to a lesser extent than in dairy cattle, especially regarding Se status in colostrum and the relationship between Se and blood parameters [12,13]. In particular, few studies have been conducted on Se in the Belgian Blue cattle breed in commercial farms, which is a predominant beef breed in Belgium. The Belgian Blue breed is particularly sensitive to trace element deficiencies such as Se because of biological (e.g., increasing nutrient demand with genetic selection and higher sensitivity to myopathy due to their body composition) and environmental factors (e.g., impoverishment of soils and forages in Belgium where the breed is mostly found) [5]. Se supplementation, monitoring and the understanding of relationships with different factors are therefore essential to prevent deficiencies and preserve health and performances of the animals on farms.

The main objectives of this study were to assess (1) the plasma Se status (during the last months of pregnancy) and colostrum Se status of properly supplemented Belgian Blue cows on commercial farms, and (2) the relationship between Se concentrations in the plasma and colostrum. Secondary objectives were to assess the relationship between plasma or colostrum Se concentration and (3) the source of dietary Se (organic vs. inorganic Se) as well as (4) selected biochemical blood parameters. We focused more deeply on colostrum selenium as the literature is still scarce compared to plasma selenium.

## 2. Materials and Methods

### 2.1. Animals

The research was conducted as an observational study using animals from commercial farms. A total of 52 pre-parturition multiparous Belgian Blue cows from 5 commercial farms in the Walloon region of Belgium (i.e., 10 or 11 cows per farm) were sampled in December 2021 and January 2022. Animal and farm selection was based on the following criterions: Belgian Blue purebred cows, voluntary participation of farmers as part of the regular veterinary monitoring in routine, clinically healthy status of the cows based on physical examination, at least 10 cows with calving within the reporting period, milking of colostrum, proper Se supplementation to pre-parturition cows and different supplementation (in the form and quantity of Se supplied) between the selected farms. All cows were housed in a stable (no access to pasture in winter) and fed different diets as described in the section below. Based on expert knowledge, the selected sample can be considered representative of Belgian farms with Belgian Blue cattle in terms of diets and management practices. No control group (e.g., farms without or with improper Se supplementation) was included in the study because the aim was to strictly consider plasma and colostrum Se behavior on farms with good dietary Se supplementation practices as recommended by animal health care professionals.

### 2.2. Diets and Se Supplementations

Diets and Se supplementations were not controlled as this is an observational study, but corresponded to the regular feeding practices on each farm. Diets fed to pregnant cows and associated nutritional values are presented in Table 1. The diets and forages quality remained stable from October 2021 (winter diets), i.e., at least 2 months before the beginning of the study. Forages produced on the farm were analyzed for nutritional values by infrared technique in a laboratory of the Requasud network (Gembloux, Belgium). The nutritional values of the concentrates and raw materials bought by the farmers were based on theoretical values from the CVB system [14,15] or on data from the suppliers in the case of compound feeds. The concentration of minerals and trace element supplements, including Se, was based on the label given by the supplier. Mineral supplements were evenly mixed into the concentrates fed to the cows. Because diets were distributed ad-libitum (min 2% left after 24 h), we considered the concentration of the nutrients inside the diet (unit/kg of dry matter (DM)). The inorganic source of Se was sodium selenite for all farms; and the organic source was selenomethionine for farms 2–5 and Se-enriched yeast for farm 1. Se contained in the forages and other ingredients apart from minerals was considered consistent between farms and negligible compared to the provided supplementation [16].

**Table 1.** Ingredients and nutritional values of the diets fed during pregnancy.

|                                                      | Farm 1 | Farm 2 | Farm 3 | Farm 4 | Farm 5 |
|------------------------------------------------------|--------|--------|--------|--------|--------|
| Ingredients (% DM)                                   |        |        |        |        |        |
| Maize silage                                         | 38.4   | 24     | 21.5   | 40.9   | 19.5   |
| Grass silage                                         | 17.8   | 48.9   | 42.7   | 22.6   | 63.5   |
| Hay                                                  | -      | -      | 10.9   | -      | -      |
| Beet pulp                                            | 34.3   | 9.7    | -      | 14     | -      |
| Straw                                                | -      | -      | -      | 4.9    | -      |
| Concentrate (incl. compound feed or raw material)    | 8.1    | 16.5   | 19.6   | 14.4   | 13.9   |
| Liquid feed                                          | -      | -      | 4      | -      | -      |
| Minerals (incl. salt)                                | 1.4    | 0.9    | 1.2    | 3.2    | 3.1    |
| Nutritional values                                   |        |        |        |        |        |
| DM (g/kg)                                            | 305    | 410    | 442    | 355    | 428    |
| CP (g/kg DM)                                         | 120    | 129    | 172    | 167    | 149    |
| NDF (g/kg DM)                                        | 398    | 475    | 408    | 459    | 452    |
| Cellulose (g/kg DM)                                  | 201    | 245    | 214    | 223    | 256    |
| DVE (g/kg DM)                                        | 68.1   | 68.3   | 76.3   | 73.8   | 69.3   |
| OEB (g/kg DM)                                        | 0      | −3.3   | 39.5   | 28.8   | 18.3   |
| NE (MJ/kg DM)                                        | 6.7    | 6.1    | 5.8    | 6.1    | 6.1    |
| Inorganic selenium (ppm) *                           | 0.4    | 0.1    | 0.33   | 0.4    | 0.9    |
| Organic selenium (ppm) **                            | 0.1    | 0.5    | 0.32   | 0.4    | 1.1    |

DM: dry matter. CP: crude protein. NDF: neutral detergent fiber. DVE: intestinal digestible protein. OEB: rumen degraded protein balance. NE: net energy calculated with the Dutch net energy evaluation (VEM) system [14]. ppm: mg/kg DM. * sodium selenite (for all farms). ** selenomethionine for farms 2–5, selenium-enriched yeast for farm 1.

### 2.3. Sample Collection and Laboratory Analyses

Blood was collected from the cows for biomarker and Se analysis between 1 and 30 days before the expected date of calving. The expected date of calving was calculated using insemination date and/or reproduction monitoring history. Samplings were done on average 16 days before the real calving date. Cows were blood sampled from tail venipuncture using a 20-gauge, 1.5-inch hypodermic needle into 8.5 mL Vacutainer tubes with no additives and 6 mL Vacutainer tubes with ethylenediamine tetra-acetic acid (K2 EDTA) anticoagulant (Becton Dickinson (BD), Franklin Lakes, NJ, USA). The samples were sent to an external laboratory (SYNLAB Belgium SRL, Heppignies, Belgium) within 24 h. The samples from the tubes with EDTA anticoagulant were centrifugated to obtain plasma. Plasma Se was analyzed by atomic absorption spectroscopy using the Varian 280Z AA instrument (Agilent Technologies Inc., Santa Clara, CA, USA). The samples from the tubes without additives were allowed to clot and serum was harvested by centrifugation. The following biomarkers were analyzed from the serum: albumin, total protein, total cholesterol, urea and vitamin B12. Albumin, total protein, total cholesterol and urea concentrations were estimated using the Abbott Alinity c system (Abbott Laboratories, Chicago, IL, USA). Vitamin B12 concentration was determined using the IMMULITE® 2000 XPi system (Siemens Healthcare GmbH, Erlangen, Germany).

Colostrum was collected into a sterile 30 mL recipient at calving after fore-stripping. Colostrum samples were directly stored at −20 °C until they were sent to the laboratory (SYNLAB Belgium SRL) at the end of the study period. Colostrum Se was analyzed by atomic absorption spectroscopy using the Varian 280Z AA instrument (Agilent Technologies Inc., Santa Clara, CA, USA).

### 2.4. Statistical Analysis

Three out of the 52 sampled cows were discarded because of missing or suspicious laboratory results (i.e., hemolysis), leaving 49 cows (between 9 and 11 cows per farm) for statistical analyses. All analyses were carried out using the software R (version 4.2.1).

Descriptive statistics (mean, median, standard deviation and range (min-max)) were carried out for serum biomarkers (including plasma Se) and colostrum Se. The plasma and colostrum Se statuses of the cows were assessed from these descriptive analyses. Graphical representations (boxplot and plot of cumulative frequency) were presented for colostrum Se.

The relationship between plasma Se and colostrum Se concentrations was assessed using Pearson correlation as well as a linear model with the farm as an additional fixed effect (i.e., to assess the relationship independently of a potential farm effect). The relationships between plasma or colostrum Se and individual serum biomarkers were assessed using the same procedure (only significant results were presented).

Dietary Se sources (organic/inorganic/total Se concentration in the diet) are quantitative variables with a limited number of values (i.e., maximum 5 levels (one per farm) as there is identical Se supplementation for all pre-parturition cows of the same farm). Given this characteristic, several complementary statistical analyses were conducted to assess their relationship with plasma or colostrum Se concentrations. First, the relationships were assessed individually using Kendall's rank correlation coefficient (commonly referred as Kendall's Tau) and analysis of variance (ANOVA). In the individual ANOVA models, plasma/colostrum Se was treated as a continuous response variable, and the dietary Se source levels were treated as a categorical effect variable. Differences between group means were assessed using pairwise Tukey's post-hoc test. Normality and homogeneity of variances were assessed a priori using the Shapiro–Wilk test and Levene's test, respectively. In addition, to better grasp the effect of forms and quantities of dietary Se together as well as the potential interaction between organic and inorganic Se supplementation, the relationship between plasma or colostrum Se concentrations and dietary Se source was analyzed using a linear model. Plasma or colostrum Se was treated as the dependent variable and organic/inorganic dietary Se and their interaction as fixed effects. Farm effect was not included as it is confounded with Se supplementation. In all analyses, probability values of $<0.05$ were considered to be statistically significant.

## 3. Results and Discussion

The present manuscript is an observational study involving properly supplemented animals. In the literature, most observational studies on Se status focused on all animals independently of their Se supplementation, also including animals with Se deficiencies. Therefore, for fairer comparisons, the results were discussed with reference to controlled trials on beef cows involving proper dietary Se supplementation.

### 3.1. Plasma and Colostrum Se Statuses in Belgian Blue Cows and Their Relationship

Descriptive statistics of plasma Se and colostrum Se are presented in Table 2. Colostrum Se distribution boxplot and cumulative frequency plot are presented in Figure 1.

**Table 2.** Descriptive statistics of plasma selenium, serum biomarkers and colostrum selenium concentrations.

|  | Median | Mean | SD | Min | Max |
|---|---|---|---|---|---|
| Plasma selenium concentration (µg/L) | 91 | 90.06 | 15.14 | 57 | 126 |
| Serum biomarker concentration |  |  |  |  |  |
| Urea (mg/dL) | 29 | 28.06 | 7.06 | 13 | 38 |
| Total cholesterol (mg/dL) | 99 | 100.24 | 20.18 | 62 | 170 |
| Total proteins (g/L) | 74 | 73.46 | 5.48 | 61 | 85 |
| Albumin (g/L) | 30 | 30.71 | 4.45 | 21 | 42 |
| B12 (ng/L) | 221 | 230.30 | 62.85 | 126 | 401 |
| Colostrum selenium concentration (µg/L) | 80 | 79.17 | 25.76 | 30 | 146 |

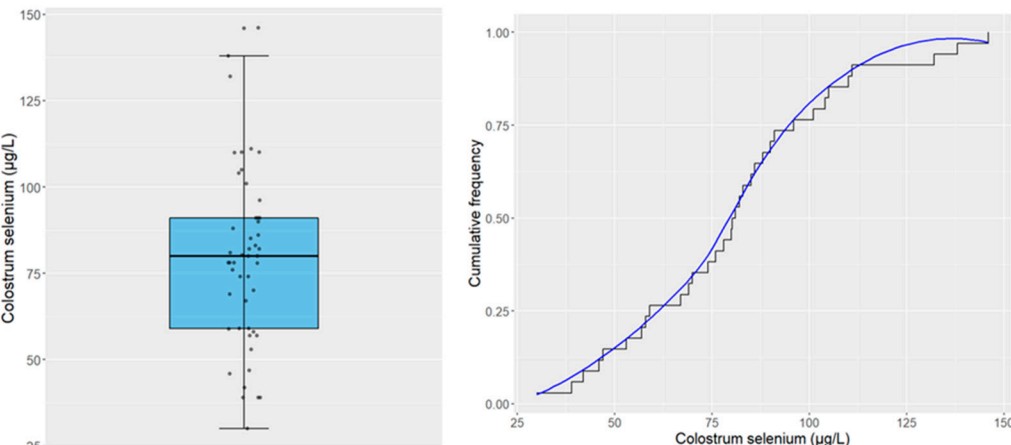

**Figure 1.** Colostrum selenium distribution (**left**); colostrum selenium cumulative frequency (the blue line is a Loess (locally estimated scatterplot smoothing) line) (**right**).

The average plasma Se concentration was 90.06 µg/L with an SD of 15.14 µg/L. Various concentrations of Se in serum/plasma in beef cows receiving supplemental organic and/or inorganic Se have been reported in the literature. For instance, Perhson et al. [17] reported mean plasma Se values of 65.5 ± 0.5 (SE) µg/L in Hereford cows in Sweden. In the USA, Muegge et al. [18] reported a mean plasma Se concentration of 125 µg/L for Angus × Simmental cows. In Belgium, Guyot et al. [5] reported mean plasma Se concentrations of 107.5 ± 15 µg/L at calving in a first study on Belgian Blue cows. In a second study with the Belgian Blue breed, Guyot et al. [19] reported mean plasma Se of 80.8 µg/L ± 2.6 (SEM) two weeks before calving. The plasma Se status of Se-supplemented Belgian Blue cows in the present study was similar to these of the two aforementioned studies involving the same breed.

In the present study, the average colostrum Se concentration was 79.17 µg/L with an SD of 25.76 µg/L, showing a high variation in individual values in non-controlled conditions. The average concentration of Se in colostrum was similar to that observed by Awadeh et al. [20] for Angus and crossbred beef cows in the USA (i.e., ~70 µg/L) and by Slavik et al. [10] for Charolais cows in the Czech Republic (i.e., 64.45 ± 7.75 µg/L). However, the observed mean concentration was higher than that reported by Muegge et al. [18] for Angus × Simmental cows in the USA (i.e., 32.5 µg/L), and by Davis et al. [21] for Angus cows in the USA (i.e., 55 ± 6.5 µg/L). Conversely, Guyot et al. [5,19] reported higher concentrations of colostrum Se in Belgian Blue cows, i.e., 150 ± 40 µg/L and 209 ± 16 (SEM) µg/L, respectively. These differences could be attributed, amongst others, to different methods of colostrum Se analysis as well as different diets and Se supplementation levels and forms.

Based on these descriptive statistics for well-supplemented cows, we would suggest marginal ranges for Se concentration in colostrum of 45–80 µg/L (i.e., 10th–50th percentile), with an acceptable minimum value of 45 µg/L, for practical recommendation in Belgian Blue cows. Further research is needed to confirm these figures, and also after comparison with populations of cows with deficient Se supplementation.

Scatter plot of plasma Se concentration vs. colostrum Se concentration is presented in Figure 2. Although data on the relationship between Se dietary intakes and Se concentrations in blood and colostrum are available (e.g., [1,19,21]), few authors directly analyzed the relationship between blood and colostrum Se concentrations using individual data. For instance, Grace et al. [22] observed a positive relationship between blood Se concentration and milk Se concentration in dairy cows, but colostrum data were not considered. On the other hand, Pavlata et al. [23] showed no significant differences in colostrum Se concentrations at widely different Se concentrations in the blood of dairy cows and hence no significant correlation between both concentrations. This is in accordance with the results of our study that demonstrated no significant relationship between Se concentrations in plasma and

in colostrum (r = 0.04; and no significant effect for the linear model). Pavlata et al. [23] argued that colostrum Se concentration would reach a plateau at relatively low blood Se concentrations and no further increases of its secretion would take place. Another possible explanation is that metabolic processes of plasma and colostrum Se differ, i.e., colostrum Se level is not the perfect image of plasma Se level. For example, colostrum Se concentration is mainly related to seleno-amino acids and protein transfers, in contrast to plasma Se concentration, which is influenced more by free inorganic Se forms [8,9]. This leads us to believe that the sole analysis of Se in the blood of the dam is not sufficient to ensure a proper Se supplementation and immunoglobulin transfer for the calf though colostrum feeding at birth, and that the Se status in colostrum is not adapted to assess the Se status of a herd.

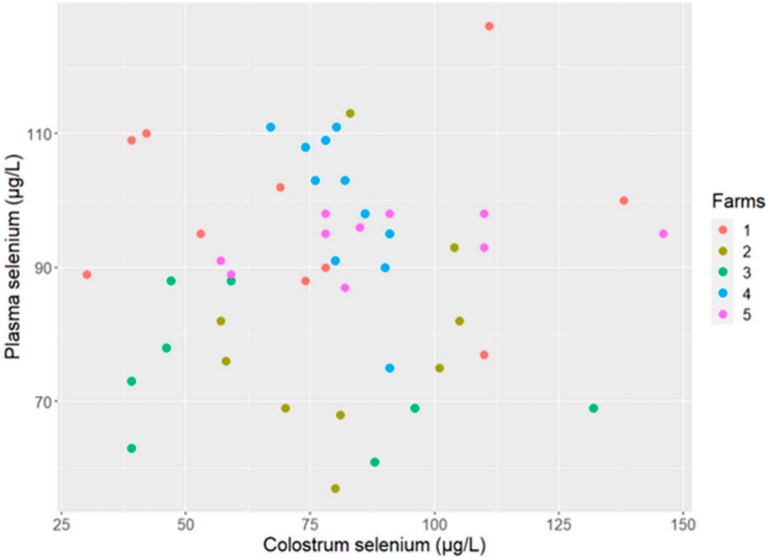

**Figure 2.** Scatter plot of plasma selenium concentration vs. colostrum selenium concentration.

### 3.2. Relationship between Plasma/Colostrum Se and the Source of Dietary Se

3.2.1. Plasma Se

The ANOVA results showed a significant difference in plasma Se concentration between cows fed different total, organic or inorganic dietary Se levels (Figure 3). The linear model indicated a significant effect for dietary inorganic Se (*p*-value = 0.016, positive estimate) and a trend for the interaction between organic and inorganic Se forms (*p*-value = 0.056, negative estimate). In addition, Kendall's Tau (T) indicated a positive relationship between plasma Se concentration and inorganic dietary Se levels (T = 0.363, *p*-value = 0.001), but no significant relationship related to total and organic dietary Se levels (T = 0.102 and −0.003, respectively). In contrast to these results, a great deal of research investigating the effect dietary Se form on blood Se concentrations showed a raise of blood Se concentration with increasing organic Se supplementation (e.g., [10,19,21]). The causes of these discrepancies can be multiple and complex, such as interaction between organic and inorganic Se forms, as shown in the results above, non-controlled diets which can among others influence Se absorption differently [9,24], physiological state of cows, high Se supplementation for some farms (i.e., between 0.8 and 2 ppm for farms 4 and 5, respectively) compared to some controlled studies in the literature (e.g., 0.1–0.5 ppm in the study of Guyot et al. [5], 0.4 ppm in the study of Guyot et al. [19]), potential confounding variables as well as complex physiological processes [8,25]. Regarding physiological processes, metabolic fates of dietary organic and inorganic Se forms differ. Organic Se not used by the liver to synthesize selenoproteins secreted in plasma is taken up by organs and tissues that have demanding protein synthesis, whereas inorganic Se such as selenite, not used immediately for selenoprotein synthesis, stays within the free pool of Se in plasma or

is excreted if in excess [8,9,26]. This might explain the observed higher Se concentration in plasma associated with higher inorganic Se supplementation.

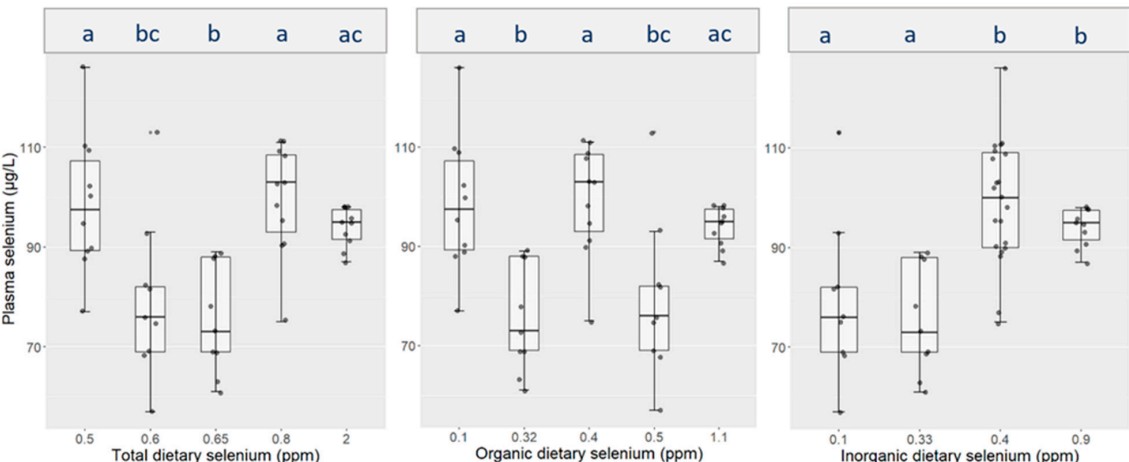

**Figure 3.** Boxplots and ANOVA results for plasma selenium concentration vs. dietary selenium levels. Total dietary selenium = organic + inorganic dietary selenium. Dietary selenium levels bearing different superscripts (in dark blue above the boxes) differ significantly at $p < 0.05$.

### 3.2.2. Colostrum Se

The ANOVA analyses for total/organic/inorganic dietary Se were all not significant, i.e., they showed no significant difference in colostrum Se concentration among the supplementation level means (Figure 4). The linear model's effects were also all not significant. However, although Kendall's Tau was not significant for total and inorganic dietary Se (T = 0.144 and 0.101, respectively), it revealed a trend for a positive relationship (T = 0.211, *p*-value = 0.05) between colostrum Se concentration and organic dietary Se concentration. Similarly to results presented for plasma Se, non-existent or weak effects could be attributed to a variety of factors such as non-controlled diets or complex physiological processes and interactions. The tendency for increased colostrum Se concentration associated with higher organic dietary Se has formerly been observed in beef cattle (e.g., [10,17,19]) and might be due to the nonspecific use by the mammary gland of selenomethionine (found in organic dietary Se forms) in place of methionine for the incorporation into milk proteins [2,27]. This may suggest, with care given the weak trend, that organic forms of Se could favor the transfer of Se from the dam to her offspring through colostrum feeding.

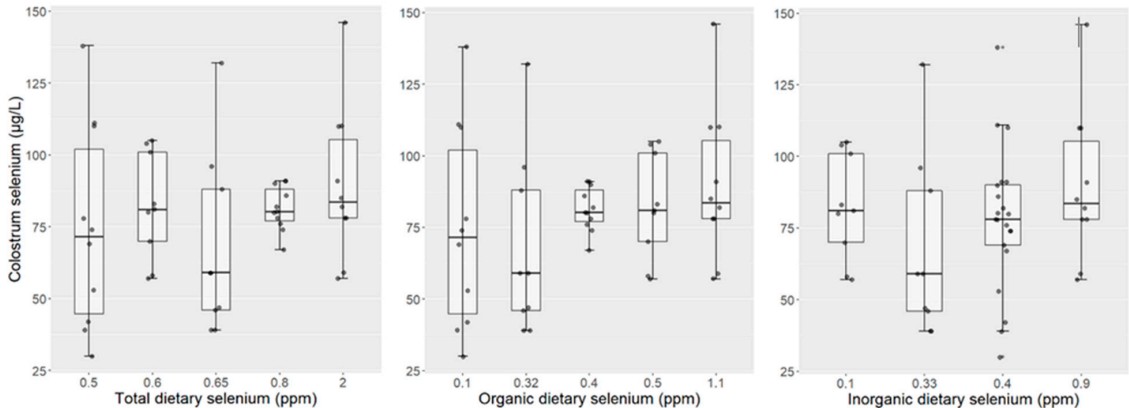

**Figure 4.** Boxplots and ANOVA results for colostrum selenium concentration vs. dietary selenium levels. Total dietary selenium = organic and inorganic dietary selenium. In the three cases, all ANOVAs were not significant (all means were equal).

*3.3. Relationship between Plasma/Colostrum Se and Serum Biomarkers*

Descriptive statistics of serum biomarkers are presented in Table 2. Correlation analyses and linear models controlling for farm effects revealed a significant negative relationship between plasma Se concentration and (1) serum urea, and (2) serum albumin concentrations (Figure 5). No significant correlations or effects in linear models were observed between plasma Se and total proteins, total cholesterol and vitamin B12. No significant relationships were observed between colostrum Se and serum biomarkers. The literature is limited with regard to this topic. As mentioned previously, in an observational study, and hence in real farming situations, many factors can be implicated and influence relationships. Serum albumin and urea concentrations are related to the nutritional and pathological status of cows. A hypothesis could be that animals with high serum albumin content (early indicator of protein status) and high serum urea content would have higher protein synthesis and a higher rate of utilization of Se in serum for incorporation in tissues with demanding protein synthesis (esp. in muscles), consequently reducing the Se pool in serum or plasma [28,29]. The causes and physiological implications of these correlations need to be investigated further.

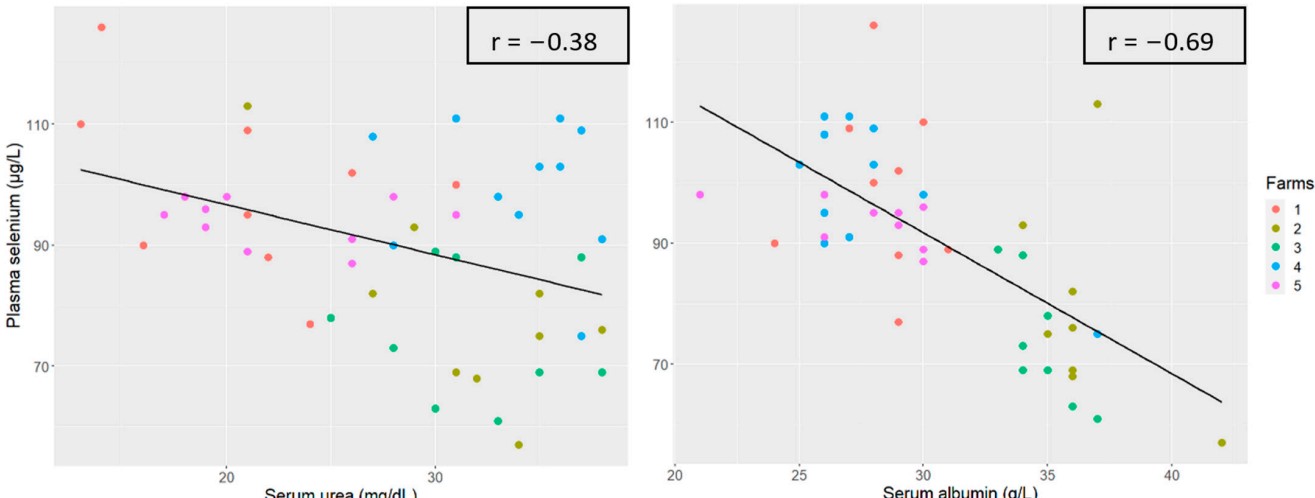

**Figure 5.** Scatter plot and correlation of plasma selenium concentration vs. serum urea concentration (**left**) and serum albumin concentration (**right**).

## 4. Conclusions

In summary, key results revealed the absence of a relationship between plasma and colostrum Se concentrations in properly supplemented Belgian Blue cows, emphasizing the role of colostrum Se as a distinct biomarker of Se metabolism in beef cows. This suggests that colostrum Se testing would be a complementary indicator for improving calf Se supplementation in addition to plasma testing in the cows. Marginal ranges of 45–80 µg/L for colostrum Se concentration in Belgian Blue cows are suggested. Our study also demonstrated the complexity of Se metabolism in non-controlled conditions on commercial farms, implying the need for further studies on this topic.

**Author Contributions:** Conceptualization, É.K., L.T. and P.D.; Methodology, P.D.; Software, P.D.; Validation, É.K., L.T. and P.D.; Formal Analysis, P.D.; Investigation, B.P., M.G., A.-S.R. and É.K.; Resources, A.-S.R.; Data Curation, P.D.; Writing—Original Draft Preparation, P.D.; Writing—Review and Editing, P.D., L.T., É.K., B.P., M.G., A.-S.R. and C.H.; Visualization, P.D.; Supervision, A.-S.R. and L.T.; Project Administration, L.T.; Funding Acquisition, L.T. All authors have read and agreed to the published version of the manuscript.

**Funding:** The RUMIN SPRL (6700 Arlon, Belgium) company partly funded laboratory analyses.

**Institutional Review Board Statement:** Ethical review and approval were waived for this study, because animal manipulations and blood samplings were performed by veterinarians as part of regular veterinary monitoring on farms (Directive 2010/63/EU).

**Informed Consent Statement:** Not applicable.

**Data Availability Statement:** The data presented in this study are available on request from the corresponding authors.

**Acknowledgments:** The authors gratefully acknowledge the farmers who kindly participated in this study.

**Conflicts of Interest:** The authors declare no conflict of interest. The funder had no role in the design of the study; in the collection, analyses, or interpretation of data; in the writing of the manuscript, or in the decision to publish the results.

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
