# Peer review of "Plasma and Colostrum Selenium Statuses of Properly Supplemented Belgian Blue Cows on Commercial Farms and Their Relationship with Sources of Dietary Selenium and Blood Biomarkers"

_2624-862X, doi:10.3390/dairy3040059_

Round 1

Reviewer 1 Report

The major suggestion is as follows;

L195 Table 2 correct decimal point from "," to "." of all data

L249-250 Figure 3 check/delete superscript "c" significant different for Total dietary+organic Se supplementation

L275-292 Suggest deleting part 3.3 Result and discussion. Part 3.3 is not suitable because the nutritional value (e.g. protein) of diets was not controlled to be similar among the farm and animals  

Reviewer 2 Report

The article is interesting and is based on sound scientific methodology including appropriate statistical treatments of the data. The results are well presented and discussed. The implications of the study have been appropriately perceived that colostrum Se testing would be a complementary indicator for improving calf Se supplementation in addition to plasma testing in the cows. Overall, the article is well written and clear for the readers and will be useful for the researchers engaged in these areas of mineral nutrition and physiology.

Reviewer 3 Report

General comments

The work shown interesting information about Se metabolism in cows. Below are some suggestions for work.

Material and methods

The authors have the values of Se present in the ingredients (maize, grass, hay)? Insert in the manuscript.

The different ingredients present in the diets (table 1) can influenced in the results of your work?  

Because the author utilized 2 source of selenium (organic and inorganic)? If utilized on source can be better? Explain in the manuscript.

Results and discussion

Figure 5 – The “r = -0.38” of correlation of first graphic is reliable?

Because inorganic Se such as selenite not used immediately for seleno-protein synthesis stays within the free pool of Se in blood or is excreted if in excess? Explain in the text.

“An hypothesis could be that animals with high serum albumin content (early indicator of protein status) and high serum urea content would have higher protein syn-285 thesis and a higher rate of utilization of Se in serum for incorporation in tissues, conse-286 quently reducing the Se pool in serum [27,28].” Explain better how the selenium is incorporated in the tissues? What tissues selenium is stored?

4. Conclusions

Based in results of work, what the level better of Se for Belgian Blue cows?

Reviewer 4 Report

The manuscript entitled Plasma and colostrum selenium statuses of properly supplemented Belgian cows on commercial farms and their relationship with source of diatery selenium and blood biomarker. Here in this article, the author assess (1) the plasma and colostrum Se statuses of properly-supplemented Belgian-Blue cows on commercial farms and (2) the relationship between Se concentrations in plasma and colostrum. Secondary objectives were to assess relationships between plasma or colostrum Se concentrations and dietary Se supplementation as well as blood biomarkers. This article is interesting, although I listed a few items below for the authors considerations. The research manuscript needs revisions and improvements.

Comment 1. How about the location of the farm, please describe it in material and methods session,

Comment 2. Table 1. Several are blank, is that 0 kg in composition?

Comment 3. Please describe more about Se metabolism in the introduction part

Comment 4. Why did correlate the colostrum Se and total protein, total cholesterol and vitamin B12?
